# Innate Immune Response Analysis in Meniscus Xenotransplantation Using Normal and Triple Knockout Jeju Native Pigs

**DOI:** 10.3390/ijms231810416

**Published:** 2022-09-08

**Authors:** Seungwon Yoon, Yunhui Min, Chungyu Park, Dahye Kim, Yunji Heo, Mangeun Kim, Eugene Son, Mrinmoy Ghosh, Young-Ok Son, Chang-Gi Hur

**Affiliations:** 1Cronex Co., Jeju-si 63078, Korea; 2Interdisciplinary Graduate Program in Advanced Convergence Technology and Science, Jeju National University, Jeju-si 63243, Korea; 3Division of Animal Genetics and Bioinformatics, The National Institute of Animal Science, RDA, Wanju 55465, Korea; 4Department of Animal Biotechnology, Faculty of Biotechnology, College of Applied Life Sciences, Jeju National University, Jeju-si 63243, Korea; 5Cronex Inc., Hwaseong 18525, Korea; 6Department of Biotechnology, School of Bio, Chemical and Processing Engineering (SBCE), Kalasalingam Academy of Research and Educational, Krishnankoil 626126, India

**Keywords:** xenotransplantation, triple knockout, meniscus, immune rejection

## Abstract

Although allogenic meniscus grafting can be immunologically safe, it causes immune rejection due to an imbalanced tissue supply between donor and recipient. Pigs are anatomically and physiologically similar to adult humans and are, therefore, considered to be advantageous xenotransplantation models. However, immune rejection caused by genetic difference damages the donor tissue and can sometimes cause sudden death. Immune rejection is caused by genes; porcine *GGTA1*, *CMAH*, and *B4GLANT2* are the most common. In this study, we evaluated immune cells infiltrating the pig meniscus transplanted subcutaneously into BALB/c mice bred for three weeks. We compared the biocompatibility of normal Jeju native black pig (JNP) meniscus with that of triple knockout (TKO) JNP meniscus (α-gal epitope, N-glycolylneuraminic acid (Neu5Gc), and Sd (a) epitope knockout using CRISPR-Cas 9). Mast cells, eosinophils, neutrophils, and macrophages were found to have infiltrated the transplant boundary in the sham (without transplantation), normal (normal JNP), and test (TKO JNP) samples after immunohistochemical analysis. When compared to normal and sham groups, TKO was lower. Cytokine levels did not differ significantly between normal and test groups. Because chronic rejection can occur after meniscus transplantation associated with immune cell infiltration, we propose studies with multiple genetic editing to prevent immune rejection.

## 1. Introduction

The meniscus, a C-shaped cartilage in the knee, acts as a shock absorbent in knee joint function. However, the meniscus can be injured by accidents or affected by aging, and an injured meniscus causes instability, pain, and cartilage degeneration in the knee joint [1]. Interestingly, the meniscus is hard to heal naturally, as blood vessels are not distributed in the area, except in the edges [2]; therefore, in some cases, it may transmit excessive load to the knee joint and cause osteoarthritis (OA) [3]. Meniscectomy is performed to treat a severely ruptured meniscus. Allogenic meniscus grafting is performed to treat damaged menisci and prevent degeneration in the knee joint [4]. From an immunological viewpoint, the allogenic meniscus can have an excellent therapeutic effect due to its relatively low immune rejection response. However, it has an issue with tissue supply due to an imbalance between the recipients and donors [5].

To overcome this imbalance, human–animal xenotransplantation was developed; in 1964, the first heart transplant utilized a chimpanzee as the donor. Nevertheless, primates are challenging to breed, and there are limitations in their use as a xenotransplantation model because of numerous ethical issues [6]. Researchers have used pigs as an animal model for primates, and they have become the most popular animal model in the xenotransplantation field. Pigs are anatomically and physiologically similar to humans, and easy to breed. They have litters and a relatively short sexual maturation and gestation period. Their organ sizes are similar to those of adult humans, making them advantageous for xenotransplantation modeling [7]. 

However, various immune rejection responses caused by genetic differences between different species are still considered obstacles to be solved. Immune rejection responses that occur within minutes or hours of xenotransplantation can damage the donor tissue and even endanger the life of recipients [8]. The most common cause of immune rejection response is the interaction between human immune systems and carbohydrate antigens on the animal cell surface [9]. Acellularization of animal tissues has been attempted as a method to remove antigens at the clinical stage for therapeutic purposes [10,11]. However, all the cells were not completely removed, and there were functional limitations. Currently, with the development of gene editing technology and cloning technology, producing animals with a specific gene deleted or inserted at the germline level (genetically modified animals) to remove immune-rejection-inducing antigens is possible. The α-gal epitope, N-glycolylneuraminic acid (Neu5Gc), and Sd (a) antigens are the most well-known carbohydrate antigens associated with immune rejection [12,13]. Humans possess natural antibodies against these antigens. When animal organs are transplanted, the immune system of recipients is activated by the interaction between antigens and antibodies, resulting in severe immune rejection [14,15,16,17].

In this study, we aimed to provide primary data for successful xenogeneic organ transplantation by investigating the level of the immune responses induced after subcutaneous transplantation of Jeju native black pig (JNP) meniscus into mice when compared with those induced after subcutaneous transplantation of triple knockout (TKO) JNP meniscus, in which three carbohydrate antigens were removed by gene editing, into mice. The interpretations from genetic editing performed in this study provide scope for further research to develop a model in which immune rejection does not occur. This would include the application of multiple forms of genetic editing.

## 2. Results

### 2.1. Histological Analysis and Meniscus Transplant Procedures

The meniscus was extracted and cut into sections from the normal and TKO JNP (Figure 1A). Under isoflurane anesthesia, the meniscus was transplanted into the dorsal back dermis of mice (Figure 1B), and the mice were bred for three weeks (Figure 1C). All of the experimental mice survived the meniscus transplant periods (Figure 1D). The normal and TKO JNP groups showed signs of recovery, such as hair growth at the transplant site. We confirmed the infiltration of cell types across the transplant border using H&E staining (Figure 2A). The TKO JNP group had fewer infiltrated cells than the normal group. Moreover, subcutaneous transplants of the normal and TKO JNP groups had excellent states of extracellular matrix and a small number of chondrocyte-like fibroblasts in the transplant center (Figure 2B).

### 2.2. Analysis of Mast Cell Distribution across the Transplant Border

The normal and TKO JNP groups had increased mast cell count across the transplant border (Figure 3A). The normal group contained approximately 302 ± 15 mast cells. Mast cells were substantially increased compared with those in the sham group (*p* < 0.0001). The TKO group also showed increased mast cell count compared to the sham group; however, when compared with the normal group, mast cell numbers were reduced by approximately 36% (*p* = 0.0233) (Figure 3B).

### 2.3. Analysis of Eosinophil and Neutrophil Distribution across the Transplant Border

The normal and TKO JNP groups showed increased levels of eosinophils and neutrophils extracellular trap (NET) across the transplant border (Figure 4A,C). The normal group contained 260 ± 25 eosinophils. The eosinophil count dramatically increased in the normal group when compared with those in the sham group (*p* < 0.0001). The TKO JNP group also showed increased eosinophil count when compared with the sham group; however, when compared with the normal group, cell count was reduced by approximately 32% (*p* = 0.0076) (Figure 4A). We determined that the normal group showed a higher NET count than the sham group by measuring CDr15 NET fluorescence intensity (*p* < 0.0001). However, NET was decreased by approximately 38% in the TKO group when compared with the normal group (*p* = 0.0462) (Figure 4D).

### 2.4. Analysis of Macrophage Distribution across the Transplant Border

The normal and TKO JNP groups showed a remarkable M1 macrophage count across the transplant border (Figure 5A). The normal group showed increased macrophage intensity when compared with the sham group (*p* = 0.0003). The TKO group showed higher macrophage intensity than the sham group; however, when compared with the normal group, it showed significantly decreased intensity (*p* = 0.0328) (Figure 5B).

## 3. Discussion

The meniscus and articular cartilage possess a crucial anatomical and functional relationship following embryological development [18]. Changes in meniscus structure affect the distribution of forces and increase the pressure load on articular cartilage, resulting in OA onset. This impact of degenerative and trauma-related meniscal injuries on the articular cartilage demonstrates the crucial relationship between the meniscus and articular cartilage.

Notably, the main component of the meniscus is water (approximately 72%), and it is composed of organic matters (approximately 28%), such as the extracellular matrix (ECM) and cells. In addition, the meniscus has reduced exposure to the recipient blood, owing to the small number of blood vessels on the outer side [1]. Therefore, unlike solid organs, the meniscus may have a relatively reduced immune response for allografts or xenografts. However, continuous exposure after transplantation can induce chronic immune rejection. Therefore, determining a method to avoid immune rejection is essential [19].

Xenotransplantation is a feasible alternative due to its high cell availability, biomaterial quality, and genetic engineering capabilities. Thus, the process of immune rejection induced by xenogeneic cartilage transplantation should be studied to develop counterstrategies. Pigs are considered to be the best animal source for xenotransplantation because they are domesticated, reproduce in large numbers, and have organs and physiologies similar to those of primates. Consequently, substantial efforts have been made to genetically modify pigs at the germline level using genetic technologies [20,21,22]. Therefore, pigs are considered the best animal source for both solid organ xenotransplantation and xenogeneic cellular therapy development [23].

To analyze the effect of meniscal transplantation on the immune system of recipients, we performed tissue staining of the mouse dermis transplanted with a normal JNP and TKO JNP meniscus. H&E staining revealed that the cells were located throughout the tissue, with various phenotypes infiltrating the transplant border. This demonstrated that matrix histoarchitecture was retained in the meniscus, with radial tie fibers present in the red–white and red zones. Through Safranin O staining, we confirmed that no cells infiltrated the center of xenograft, and it remained intact. Further analysis by Alcian blue, Congo red, CDr15, and F4/80 antibody staining techniques confirmed that the infiltrated cells were innate immune cells, such as mast cells, eosinophils, neutrophils, and macrophages. Mast cells and eosinophils play an essential role in the allergic inflammatory response and are known to initiate the production of cytokines in response to infection or tissue damage [24,25]. Neutrophils are the most abundant circulating leukocytes. They were the first cell population that infiltrated into transplants, thereby causing NETs [26]. NETs are recognized by macrophages as damage-associated molecular patterns (DAMPs) and are known to act as the initiator of IL-1β production and infection signals [26].

Macrophages activated by the inflammatory response differentiate into M1 cells and produce pro-inflammatory cytokines, such as IL-1β, IL-6, IL-12, IL-23, and TNF-α. These pro-inflammatory cytokines aggravate immune rejection by activating adaptive immune cells [27]. However, pro-inflammatory cytokine concentrations in the blood of normal and TKO JNP group, when compared with the sham group, showed insignificant differences (Appendix A). Stapleton et al. demonstrated that many macrophages and fibroblast-like cells were identified at the transplant boundary in a mouse transplanted subcutaneously with acellularized pig meniscus [28]. 

However, the infiltration of adaptive immune cells, such as CD3- and CD4-positive cells, was limited. Although TKO JNP meniscus affects innate immune cell infiltration, it is presumed that the effect on systemic inflammatory responses, such as pro-inflammatory cytokine production and adaptive immune cell activation, is insignificant. The cells and ECM components of TKO JNP meniscus act as antigens. However, it is assumed that the gel-like meniscus matrix properties minimize antigen exposure to the cell surface and thereby reduce the interaction with the immune system of recipients. [29]. Wang et al. reported that α-Gal epitope, Neu5Gc, and Sd (a) antigens are expressed in various organs of normal pigs, such as the heart, liver, lung, kidney, spleen, and pancreas [30]. Therefore, we analyzed the effects of carbohydrate antigen-removed meniscal transplants on the immune system of recipients. The meniscus of pigs from which the three antigens were removed showed decreased infiltration of innate immune cells compared to the normal meniscus. 

Furthermore, there was a substantial difference in immune cell infiltration between the TKO JNP and normal groups. Thus, TKO pigs may be an ideal source of organs for future clinical xenotransplantation. However, they are less suitable for preclinical studies in non-human primate models [16,31,32]. In this study, no remarkable difference in blood cytokine levels was associated with systemic immune rejection, suggesting that TKO-sensitization would have no detrimental effects on subsequent organ allotransplantation. This observation is consistent with the majority of previous research findings, in which non-human primates or humans were sensitized to pig allergens [33].

Unlike humans, apes, monkeys, and rodents do not have natural antibodies against the three carbohydrate antigens; therefore, removing the three carbohydrate antigens was presumed to reduce the non-specific binding of innate immune cells. However, as there were various infiltrated cells across the transplant boundary, unknown epitopes that affect immune cell recruitment existed. These could cause various side effects during long-term survival after meniscal transplantation.

## 4. Materials and Methods

### 4.1. Animals and Materials

The inner and lateral menisci of normal JNPs and TKO JNPs (GGTA1, CMAH, and B4GALNT2 knockout) were provided from CRONEX Co., Ltd. (Jeju-si 63078, Korea). Five-week-old male BALB/c mice (*n* = 11), purchased from Samtaco Bio Korea Co., Ltd. (Gyeonggi 18100, Korea), were used as recipients. Mayer’s hematoxylin (Agilent Dako, Santa Clara, CA, USA), eosin Y (Sigma-Aldrich, St. Louis, MO, USA), Fast green (Sigma-Aldrich, St. Louis, MO, USA), Safranin O (Sigma-Aldrich, St. Louis, MO, USA), Congo red (Sigma-Aldrich, St. Louis, MO, USA), sodium hydroxide (Sigma-Aldrich, St. Louis, MO, USA), Alcian blue (Sigma-Aldrich, St. Louis, MO, USA), and nuclear fast red (Sigma-Aldrich, St. Louis, MO, USA) were used for histochemical analysis. Recombinant anti-F4/80 antibody (ab111101, Abcam, Waltham, MA, USA), goat antirabbit IgG Alexa Fluor plus 594 (A32740, Invitrogen, Waltham, MA, USA), gold antifade mountant with 4′,6-diamidino-2-phenylindole (DAPI) (Invitrogen, Waltham, MA, USA), and CDr15 were provided by Pohang University of Science and Technology, Gyeongsangbuk-do, South Korea, and were used for immunohistochemical and immunofluorescence analyses.

### 4.2. Meniscus Xenotransplantation

The meniscus extracted from normal and TKO JNPs was washed with 70% ethanol and phosphate-buffered saline (PBS) containing 1% antibiotics. Each meniscus was cut into sections of approximately 50 mm^3^ and immersed in PBS before transplantation. Recipient mice (*n* = 11) were divided into three groups: a sham group without xenotransplantation (*n* = 3); a normal group transplanted with the normal pig meniscus (*n* = 4); and a test group transplanted with the TKO pig meniscus (*n* = 4). Under respiratory anesthesia with isoflurane, we made a posterior incision using surgical tools and transplanted a piece of the meniscus into the dorsal dermis and sutured it. All three groups of mice were bred for three weeks in a breeding room at the Experimental Animal Center of Jeju National University for further analysis. All animal experiments were performed according to procedures approved by the Animal Welfare Committee of Jeju National University (2022-0035) and CRONEX Co., Ltd., Jeju, Korea (CRONEX-IACUC:202101008).

### 4.3. Histologic Analysis

After extracting the mouse dorsal dermis transplanted with pig meniscus, the tissues were fixed with 4% paraformaldehyde at 4 °C for one day. The fixed tissues were decalcified by immersion in a 0.5 M ethylenediaminetetraacetic acid (EDTA) solution (pH 7.4) for approximately seven days. The paraffin-embedded dermis was sliced to a thickness of 5 μm, transferred to slides, dewaxed with xylene, and rehydrated with gradient ethanol. Each slide was stained with H&E, Safranin O, Alcian blue, and Congo red. All stained slides were analyzed to determine the infiltration of cell types toward the transplant border in each group, using a LEICA DM 2500 microscope (Leica Camera Inc., Allendale, NJ, USA).

### 4.4. Immunofluorescence Microscopy

After dewaxing with xylene and rehydrating with gradient ethanol, the slides were treated with a peroxidase block solution (Golden Bridge International, Inc., City of Industry, CA, USA) for peroxidase inhibition. After antigen retrieval with 0.05% trypsin solution (Gibco, Waltham, MA, USA), blocking was performed using 1% bovine serum albumin. Slides were stained with 10 μM CDr15 to analyze the distribution of NET. For macrophage distribution analysis, slides were sequentially stained with the diluted recombinant anti-F4/80 antibody (dilution 1:500) and diluted goat antirabbit IgG Alexa Fluor plus 594 (concentration 1:1000) as the secondary antibody. All slides were stained with gold antifade mountain stain with DAPI for nuclear staining to compare the immune response between each group. Slide images taken with a Cytation 5 Cell Imaging Multimode Reader (BioTek, Santa Clara, CA, USA) at the Bio-Health Materials Core Facility, Jeju National University, and were analyzed for fluorescence intensity using ImageJ software (NIH, Bethesda, MD, USA) (probe CDr15 detection wavelength: Ex/Em = 560 nm/730 nm).

### 4.5. Statistical Analysis

The mean and standard deviation of all cell counts and immunofluorescence intensity data were calculated using IBM SPSS Statistics 24 (IBM Corp., Armonk, NY, USA). Data from the normal JNP and TKO JNP groups were compared using the two-tailed independent *t*-test. A *p* value < 0.05 was considered a significant difference.

## 5. Conclusions

We analyzed the effect of the immune response induced in the xenotransplantation of meniscus and removal of three well-known carbohydrate antigens as immune rejection factors. Subcutaneous transplantation of the meniscus caused the infiltration of many innate immune cells into the transplant border. However, a systemic inflammatory response was not observed. The removal of the three carbohydrate antigens by gene editing resulted in relatively low infiltration of innate immune cells. However, side effects, such as chronic rejection, can occur during long-term transplantation of residual immune cells. Eventually, to increase the stability of meniscus xenotransplantation, multiple editing or removal of factors that induce immune rejection, such as swine leukocyte antigen-1 and endogenous retrovirus, should also be considered. We presume that our study provides a model that addresses preclinical testing requirements. Although the current study has limitations, which include a relatively small number of animals that is insufficient to provide statistical power to the results and an in-depth immunological investigation, we believe that it still provides a reference for the stated hypothesis question of an appropriate animal model for the assessment of menisci xenografts.

## Figures and Tables

**Figure 1 ijms-23-10416-f001:**
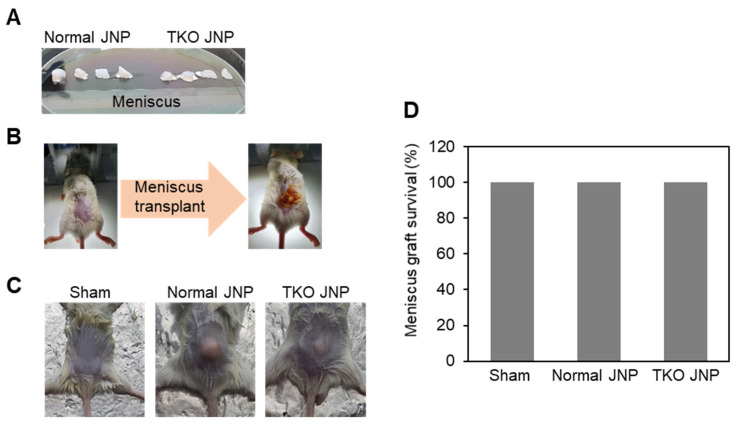
Procedures for meniscus transplantation of normal/TKO JNP and graft survival rate. (**A**) The meniscus from the normal and TKO JNP groups was removed and divided into sections measuring about 50 mm^3^. (**B**) Mice were given isoflurane anesthesia while the meniscus was implanted in the dorsal back dermis. (**C**) After three weeks of breeding, the mice were euthanized for additional histological analysis. (**D**) Meniscus graft survival rate was illustrated.

**Figure 2 ijms-23-10416-f002:**
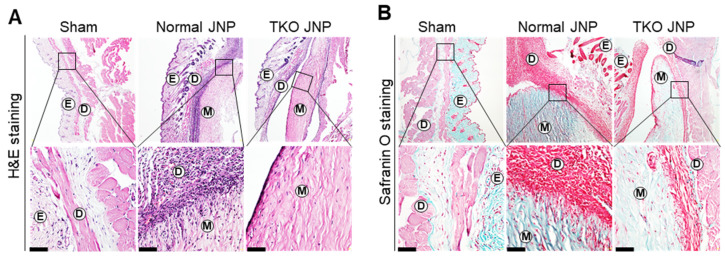
Histological analysis of infiltrated cells in the sham, normal, and TKO JNP groups. The meniscus is extracted from the normal and TKO JNP groups and was cut into sections of approximately 50 mm^3^. The meniscus was transplanted in the dorsal back dermis of mice. The mice were bred for about three weeks. The mouse dermis containing the pig meniscus and sham was examined by H&E (**A**) and Safranin O staining (**B**). E: epidermis; D: dermis; M: meniscus. Magnification (upper panel: 100X, lower panel: 400X). Scale bars: 50 μm.

**Figure 3 ijms-23-10416-f003:**
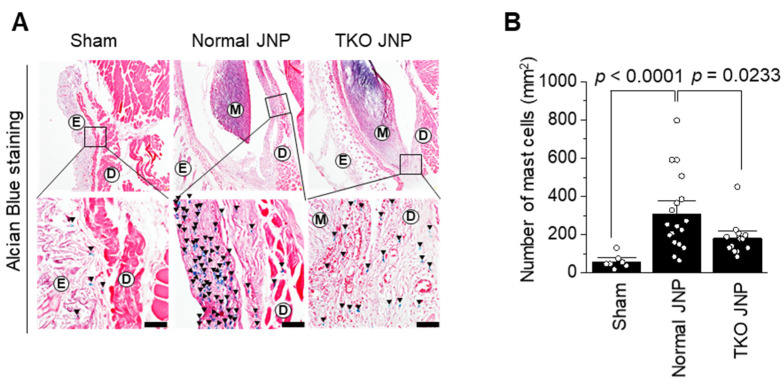
Immunohistochemical analysis of mast cell activation in the sham, normal and TKO JNP groups. The distribution of mast cells was partially confirmed by Alcian blue staining (**A**), and the quantification of mast cells in each group is represented (**B**). In each field, mast cell count for the sham (n, observed field = 7), normal JNP (n, observed field = 18), and TKO JNP (n, observed field = 13) were determined. A two-tailed independent *t*-test was used for the statistical analysis. E: epidermis; D: dermis; M: meniscus. Magnification (upper panel: 100X, lower panel: 400X). Scale bars: 50 μm.

**Figure 4 ijms-23-10416-f004:**
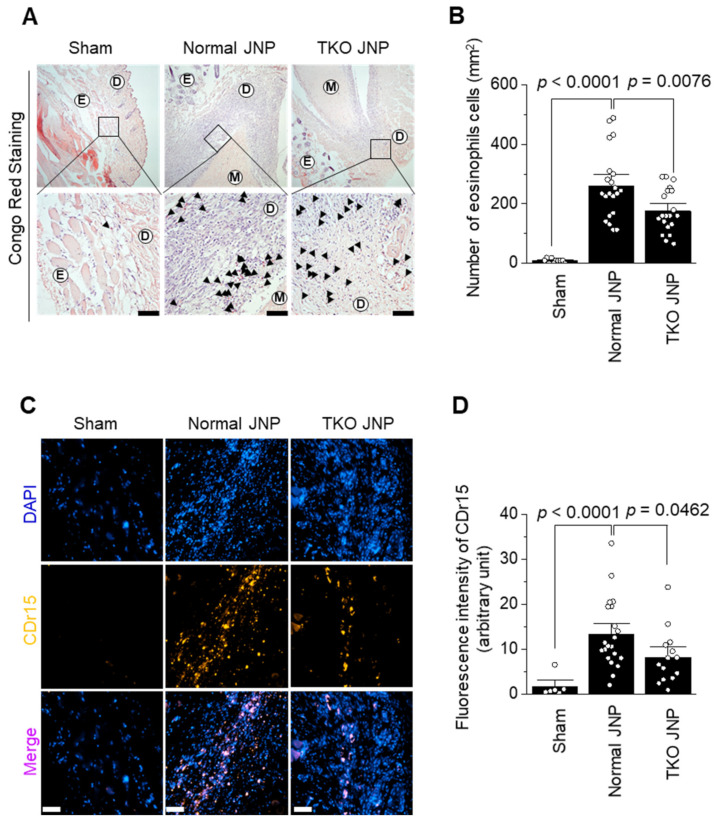
Immunohistochemical analysis of eosinophil and extracellular trap (NET) infiltration in the sham, normal, and TKO JNP groups. The distribution of eosinophils and NET was analyzed by the Congo red (**A**) and CDr15 (**C**) staining. The quantification of eosinophils (**B**) and NET (**D**) is represented. In each field, eosinophil count for the sham (*n* = 10), normal (*n* = 21), and TKO JNP (*n* = 20) groups; and NET count for the sham (*n* = 6), normal (*n* = 21), and TKO JNP (*n* = 14) groups were determined. A two-tailed independent *t*-test was used for statistical analysis. E: epidermis; D: dermis; M: meniscus. Magnification of A (upper panel: 100X, lower panel: 400X) and C (200X). Scale bars: 50 μm. DAPI (blue for nucleus), CDr15 (Yellow for neutrophil extracellular trap), Merge (DAPI + CDr15).

**Figure 5 ijms-23-10416-f005:**
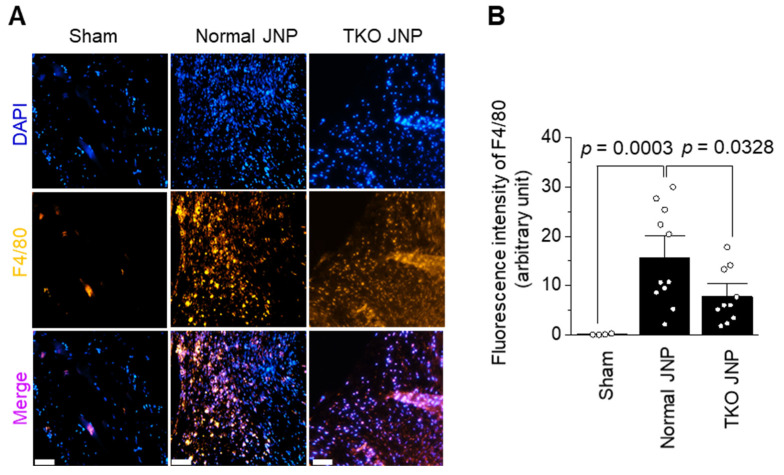
Immunohistochemical analysis of macrophage infiltration in the sham, normal, and TKO JNP groups. The distribution of macrophages was analyzed by anti-F4/80 antibody staining (**A**), and the fluorescence intensity of F4/80 was represented in (**B**). In each field, macrophage counts for the sham (*n* = 4), normal (*n* = 11), and TKO JNP group (*n* = 10) were determined. A two-tailed independent *t*-test was used for significant analysis. Magnification: 200X. Scale bars: 50 μm. DAPI (blue for nucleus), F4/80 (Yellow for macrophage), Merge (DAPI + F4/80).

## Data Availability

Not applicable.

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
