# Peer review of "Innate Immune Response Analysis in Meniscus Xenotransplantation Using Normal and Triple Knockout Jeju Native Pigs"

_ijms, 2022, doi:10.3390/ijms231810416_

Round 1

Reviewer 1 Report

The authors have presented limited data from meniscus transplantation from Normal and Triple Knockout Jeju Native Pigs to rodents. The mere presence of cells in the transplanted tissue does not mean that much overall. It would be better to follow the recipients for a longer time and analyze more in depth the effect on the recipient.

Additionally, there are several questions which need to be address.

Why did the authors choose BALB/c mice, over the more traditional B6?

Why was the subQ route selected?

What is the time point at which the analysis was performed, post-surgery?

The authors mention breeding the mice after surgery, which makes no sense to the reviewer.

Did the mice receive any antibiotics or pain relivers after surgery?

Do you have any pictures of the tissue or the surgical site that can be included in the publication?

Can you provide more information on the transgenic pigs? Are they physiologically normal? Is the meniscus harvested from the knockouts, structurally similar tot eh normal pigs? Are they fertile?

What was the cellularity of the transplanted tissue? Can you detect any pig cells in the transplant recipients? Can you use a pan-pig antibody to detect pig cells?

In the Figures the authors use the term Normal, and, in the text, it is written as Control. Keep it consistent so it is easier to read.

The CDr15 staining detects neutrophil traps, not actual neutrophils. Is that correct? If so, this needs to be addressed in the paper.

Did the authors look at the various immune cell subsets in mice? The draining lymph nodes would have been an excellent option to analyze, so would the spleen.

Vehicle is a very odd term to use for surgical groups, sham surgery is a better term.

For the vehicle group, did the mice receive, any tissue or just PBS?

For both the eosinophil and neutrophil figures, the authors mention a 60% reduction in the knockout pig tissue as compared to the normal tissue. However, the figures do not support this.

The figures show that the infiltration was reduced to 60%, i.e., the reduction was only 40%.

The authors should give values for the knockout groups if they see a 60% reduction.

Authors were unable to detect any cytokines in the recipient blood, which is concerning. What assay was used? More details are needed. Did the authors analyze IL1 or IL27?

Author Response

Reviewer #1:

The authors have presented limited data from meniscus transplantation from Normal and Triple Knockout Jeju Native Pigs to rodents. The mere presence of cells in the transplanted tissue does not mean that much overall. It would be better to follow the recipients for a longer time and analyze more in depth the effect on the recipient.

Additionally, there are several questions which need to be address.

Here is the list of my comments and opinions:

Comment 1.

Why did the authors choose BALB/c mice, over the more traditional B6?

[Response]

BALB/c mice tend to exhibit more robust humoral responses compared to C57BL/6 mice [1][2]. In a previous study, most infiltrating cells around the implant were macrophages or Fibroblast-like cells in decellularized pig meniscus transplanted subcutaneously. Therefore, we selected BALB/C mice for the innate cell response in the meniscus transplant experiment.

Reference

[1] Bleul, T., et al., Different Innate Immune Responses in BALB/c and C57BL/6 Strains following Corneal Transplantation. J Innate Immun, 2021. 13(1): p. 49-59.

[2] Stapleton, T.W., et al., Investigation of the regenerative capacity of an acellular porcine medial meniscus for tissue engineering applications. Tissue Eng Part A, 2011. 17(1-2): p. 231-42.

Comment 2.

Why was the subQ route selected?

[Response]

In the dermis, there are various immune cells, blood vessels, and lymph nodes [3]. Pig meniscus was difficult to orthotopic transplant to mouse knee joint due to size. Therefore, subcutaneous transplantation was selected to expose the recipient's immune system to xenografts effectively.

Reference

[3] Nguyen, A.V. and A.M. Soulika, The Dynamics of the Skin's Immune System. Int J Mol Sci, 2019. 20(8).

Comment 3.

What is the time point at which the analysis was performed, post-surgery?

[Response]

We performed the histological analysis three weeks after post-surgery. Please see our Material and methods section (page 8, lines 254~256).

Comment 4.

The authors mention breeding the mice after surgery, which makes no sense to the reviewer.

[Response]

We keep the mice after surgery during experiment periods at the animal facility. The sentence looks like no problem.

Comment 5.

Did the mice receive any antibiotics or pain relivers after surgery?

[Response]

We did not apply any antibiotics or pain relievers during the experiment.

Comment 6.

Do you have any pictures of the tissue or the surgical site that can be included in the publication?

[Response]

We presented pictures of the meniscus tissue and surgery mice in Figure 1.

Comment 7.

Can you provide more information on the transgenic pigs? Are they physiologically normal? Is the meniscus harvested from the knockouts, structurally similar to the normal pigs? Are they fertile?

[Response]

Using CRISPR-Cas 9, triple knockout (TKO) JNPs for the -gal epitope, N-glycolylneuraminic acid (Neu5Gc), and Sd(a) epitope were produced. The TKO or regular JNPs were used to harvest the meniscus. The meniscus of the TKO JNP had a similar structural makeup to that of a typical pig. Our group has recently produced TKO JNP and is investigating various physical characteristics using various organ xenotransplantation experiments. However, we have not yet focused on fertility.

Comment 8.

What was the cellularity of the transplanted tissue? Can you detect any pig cells in the transplant recipients? Can you use a pan-pig antibody to detect pig cells?

[Response]

The meniscus was used as the transplanted tissue. Fibrochondrocytes make up the majority of the meniscal tissue. They function like fibroblasts and chondrocytes and produce the ECM in the meniscus [4]. We hypothesized that the pig's meniscus matrix's gel-like quality shields fibrochondrocyte antigens from the recipient's immune system [5]. We haven't yet observed a suitable antibody to bind porcine fibrochondrocytes.

Reference

[4] Twomey-Kozak, J. and C.T. Jayasuriya, Meniscus Repair and Regeneration: A Systematic Review from a Basic and Translational Science Perspective. Clin Sports Med, 2020. 39(1): p. 125-163.

[5] M. Ochi, Y.I., O. Ishida, and M. Akiyama Cellular and humoral immune responses after fresh meniscal allogragfts in mice. Arch Orthop Trauma Surg 1993. 112: p. 163-166.

Comment 9.

In the Figures the authors use the term Normal, and, in the text, it is written as Control. Keep it consistent so it is easier to read.

[Response]

We thank reviewer #1’ comment. Following the reviewer’s comment, we edited the “control” to the “normal” in the text.

Comment 10.

The CDr15 staining detects neutrophil traps, not actual neutrophils. Is that correct? If so, this needs to be addressed in the paper.

[Response]

The reviewer’s comment is right. The CDr15 staining detects neutrophil extracellular traps (NET). We have edited accordingly in the text.

Comment 11.

Did the authors look at the various immune cell subsets in mice? The draining lymph nodes would have been an excellent option to analyze, so would the spleen.

[Response]

Consider that the ultimate goal of this study is to create JNPs with 6 genes knocked out. Thank you so much for your insight, and we appreciate you, suggesting a further option that will undoubtedly aid us in our next study. However, we did not look at other immune cell subsets in this study.

Comment 12.

Vehicle is a very odd term to use for surgical groups, sham surgery is a better term.

For the vehicle group, did the mice receive, any tissue or just PBS?

[Response]

Thank reviewer #1 comments. We have edited “Vehicle” to “Sham” in the text and figures.

We made a dorsal incision dermis and sutured it without transplanting any tissue for the Sham operation.

Comment 13.

For both the eosinophil and neutrophil figures, the authors mention a 60% reduction in the knockout pig tissue as compared to the normal tissue. However, the figures do not support this.

[Response]

We appreciate reviewer #1 comments. We made a mistake in the calculation. The eosinophil reduction rate was 32%, and the neutrophil reduction rate was 38%. We have recalculated all reduction rates and edited them in the text.

Comment 14.

The figures show that the infiltration was reduced to 60%, i.e., the reduction was only 40%. [Response]

The reviewer #1 comment is correct. Continuing reviewer #1 comment 13, we have revised the text. 

Comment 15.

The authors should give values for the knockout groups if they see a 60% reduction.

[Response]

We already revised this issue in Comment 13.

Comment 16.

Authors were unable to detect any cytokines in the recipient blood, which is concerning. What assay was used? More details are needed. Did the authors analyze IL1 or IL27?

[Response]

We used ELISA methods to detect cytokines (IL-4, IL-6, IL-10, IFN-r, TNF-a). Mouse IL-4 ELISA kit (Cat.no BMS613, Invitrogen); assay range: 3.9 - 250 pg/mL [6], IFN gamma Mouse ELISA kit (Cat.no BMS606-2, Invitrogen); assay range: 15.6 - 1,000pg/mL [7], Mouse IL-6 ELISA kit (Cat.no BMS603-2, Invitrogen); assay range: 31.3 - 2,000 pg/mL[8], Mouse IL-10 Uncoated ELISA (Cat.no 88-7105, Invitrogen); assay range: 32 - 4,000pg/mL[9], Mouse IL-17A ELISA kit (Cat.no BMS6001, Invitrogen); assay range: 7.8 - 500 pg/mL[10], Mouse TNF alpha ELISA (Cat.no BMS607-3, Invitrogen); assay range: 31.3 - 2,000 pg/mL[11] are very specific antibody for cytokines and is used worldwide. Scientists are consider systemic immune responsive over 100 pg/mL of immune responsive cytokines [12]. Therefore we assumed that the immune response-related cytokines had not arisen in our meniscus xenograft experiments. We did not analyze the IL1 and IL27 in this study.

Reference

[6] Chen, Z., et al., Immunosuppressive effect of sinomenine in an allergic rhinitis mouse model. Exp Ther Med, 2017. 13(5): p. 2405-2410.

[7] Yang, S., et al., Dimethyl itaconate inhibits LPSinduced microglia inflammation and inflammasomemediated pyroptosis via inducing autophagy and regulating the Nrf2/HO1 signaling pathway. Mol Med Rep, 2021. 24(3).

[8] Xue, L., et al., Inhibitory effects of methamphetamine on mast cell activation and cytokine/chemokine production stimulated by lipopolysaccharide in C57BL/6J mice. Exp Ther Med, 2018. 15(4): p. 3544-3550.

[9] Zhi, Y.K., et al., Sinomenine inhibits macrophage M1 polarization by downregulating alpha7nAChR via a feedback pathway of alpha7nAChR/ERK/Egr-1. Phytomedicine, 2022. 100: p. 154050.

[10] Elashiry, M., et al., Dendritic cell derived exosomes loaded with immunoregulatory cargo reprogram local immune responses and inhibit degenerative bone disease in vivo. J Extracell Vesicles, 2020. 9(1): p. 1795362.

[11] Wu, J., et al., Study of immune responses in mice to oral administration of Flor.Essence. Mol Clin Oncol, 2020. 12(6): p. 533-540.

[12] Song, J.M., et al., Oral Administration of Porphyromonas gingivalis, a Major Pathogen of Chronic Periodontitis, Promotes Resistance to Paclitaxel in Mouse Xenografts of Oral Squamous Cell Carcinoma. Int J Mol Sci, 2019. 20(10).

Reviewer 2 Report

From my point of view, the paper can be published in IJMS if the authors could address my major and minor concerns as follow:

Major:

1. "Immune rejection analysis" is mentioned in the title. Immune rejection in transplantation has been reported also involving adaptive immune response that is primarily comprised of T and B cell. If the author also add more data for these marker for example CD3 T cell and CD20 for B cell in the IHC experiment, it would be more informative. If not, the author may consider to revised the title become more specific.

2. Authors should also show the macroscopic photo of transplantation.

3. Authors should also present the graft survival data.

Minor:

1. The magnification for the microscopic figure should be mentioned in figure legend.

2. Please describe more how to detect mast cells with only HE staining?

3. In the results, "The control group contained approximately 350 mast cells". Authors should mention SD+/-mean, not approximately. Please be specific.

Author Response

Reviewer #2:

From my point of view, the paper can be published in IJMS if the authors could address my major and minor concerns as follow:

Major:

Comment 1.

"Immune rejection analysis" is mentioned in the title. Immune rejection in transplantation has been reported also involving adaptive immune response that is primarily comprised of T and B cell. If the author also add more data for these marker for example CD3 T cell and CD20 for B cell in the IHC experiment, it would be more informative. If not, the author may consider to revised the title become more specific. 

[Response]

We acknowledge the reviewer #2's input. Because we didn't notice any systemic immune response in the recipient's cytokine levels, we chose not to analyze the adaptive immune response. But the criticism from reviewer #2 is fair. Our manuscript's title has been changed from "Immune rejection analysis in meniscus xenotransplantation using normal and triple knockout Jeju Native Pigs" to "Innate immune response analysis in meniscus xenotransplantation using normal and triple knockout Jeju Native Pigs" in order to be more accurate.

Comment 2.

Authors should also show the macroscopic photo of transplantation.

[Response]

We have represented this in Figure 1A-C.

Comment 3.

Authors should also present the graft survival data.

[Response]

We have represented this in Figure 1D.

Minor:

Comment 4.

The magnification for the microscopic figure should be mentioned in figure legend.

[Response]

We have mentioned the magnification of all microscopic figures in the legend.

Comment 5.

Please describe more how to detect mast cells with only HE staining?

[Response]

We detected mast cells by Alcian Blue staining, not H&E staining.

Comment 6.

In the results, "The control group contained approximately 350 mast cells". Authors should mention SD+/-mean, not approximately. Please be specific.

[Response]

We have edited it accordingly (page 3, line 110 and page 4, line126).

Round 2

Reviewer 1 Report

The authors have addressed all the Reviewer comments and the paper can be accepted for publication in its current format.

 Just one note: When the authors say “the mice were bred for three weeks” it sounds like you mated a male and female mouse after surgery to produce off spring. That is the reason it makes no sense to this reviewer.

Reviewer 2 Report

The authors have addressed most of my concerns.